# China's Incentives and Efforts against IUU Fishing in the South China Sea

Chengyong Yu [1] and Yen-Chiang Chang [2,*]

1    School of Law, Shandong University, Qingdao 266200, China
2    School of Law, Dalian Maritime University, Dalian 116026, China
*    Correspondence: ycchang@dlmu.edu.cn

**Abstract:** Illegal, unreported, and unregulated (IUU) fishing is a massive problem that poses a significant threat to the sustainability of marine ecosystems and the livelihoods of millions of people who depend on fishing for their food and income. Many issues have emerged, such as declining fishery resources, regional fishery incidents, political impacts, and disputes over sovereignty, which all have mutual and complicated effects on IUU fishing, eventually hindering the sustainability of marine fisheries. In this situation, the People's Republic of China (hereinafter referred to as China) has tried to undertake some efforts to combat IUU fishing over the past few years using domestic regulation and international cooperation, especially in the South China Sea. This article discusses the seriousness of IUU fishing; examines the causes of IUU fishing in the South China Sea; analyzes why frequent fishing conflicts have increased in the South China Sea; identifies what IUU fishing is, based on its definition in the International Plan of Action to Prevent, Deter and Eliminate Illegal, Unreported and Unregulated Fishing (IPOA-IUU) and in other countries; and examines what actions have been undertaken to prevent IUU fishing in China from international and national perspectives. By analyzing the cause of IUU fishing, identifying its scope and nature, and demonstrating China's position on it, this study aims to prove that China has taken some positive measures to combat IUU fishing in the South China Sea. To promote the sustainable development of fisheries in the South China Sea, uniting China and other South China Sea states against IUU fishing could be an efficient way in the future.

**Keywords:** South China Sea; illegal; unreported and unregulated (IUU) fishing; maritime Silk Road; Agreement of Port State Measures (PSMA)

## 1. Introduction

Aside from maritime disputes and growing state competition, states' interests in the sea can be generally summarized using the "three Ps": politics, petroleum, and protein (fish). Fisheries are the main source of protein and play a vital role in providing food and ensuring economic development. Utilizing fishery resources to their fullest potential has led to overfishing becoming a major problem globally, largely due to poor fishery management and IUU fishing. The results suggest that a country's risk of IUU fishing is positively related to the number of commercially significant species found within its territorial waters and its proximity to known ports of convenience [1]. The South China Sea is a marginal sea of the Western Pacific Ocean covering approximately 3.5 million km$^2$. Most coastal states in the South China Sea are developing or underdeveloped, with abundant fishery resources and important sites for the coastal aquaculture industry. The South China Sea is a vital region for the fishing industry, providing a major source of food and livelihood for the coastal communities in the area. However, in recent years, the issue of IUU fishing has become a significant problem in the South China Sea, leading to the depletion of fish stocks and damage to the marine ecosystem.

IUU fishing harms a country's economy, environment, and marine protected areas, causing a loss of economic income, environmental damage, and decline in fish stocks. IUU

fishing disrupts the sustainability of marine ecosystems and is a major threat to a country's sustainable fishery production. From an economic point of view, IUU fishing undermines the efforts of legitimate fishing industries and results in significant economic losses for coastal communities and countries [2]. From an environmental point of view, the use of illegal fishing equipment, such as mesh, fixed lines, and explosives, severely damages the marine ecosystem. Therefore, the study of combating IUU fishing in the South China Sea could have several positive impacts on regional sustainability, including the preservation of marine biodiversity, promotion of sustainable fishery-management practices, regional cooperation, and protection of food security.

## 2. Causes of IUU Fishing in the South China Sea

### 2.1. Sharp Decline of Fisheries Resources and Overfishing in the South China Sea

The South China Sea is a habitat for some of the world's richest reef systems, with more than 3000 fish species, accounting for approximately 12 percent of the total number of fish caught worldwide [3]. However, the fisheries in this region are in severe danger. Since the late 20th century, the fishery resources in the South China Sea have gradually shown a trend of rapid decline. States around the South China Sea generally have long coastlines, which strongly depend on the fishery industry. To pursue the maximum benefits of marine economics in the short term, coastal states are inclined to over-exploit fishery resources in this area. With the development of science and technology, more advanced and efficient fishing techniques have exacerbated overfishing in the South China Sea [4]. In addition, because the South China Sea is a typical semi-enclosed sea that relies mainly on the marine environment, the regeneration ability of its fishery resources is limited. As the Food and Agriculture Organization of the United Nations (FAO) published in "The State of World Fisheries and Aquaculture 2018 (SOFIA)," most fish populations in the mid-western Pacific, especially in the western part of the South China Sea, are over-fished [5].

### 2.2. Regional Fishery Conflicts and International Political Impacts

Regional fishery conflicts in the South China Sea are complex and multifaceted, with issues ranging from maritime disputes to overfishing and IUU fishing. In the South China Sea, fishery conflicts among fishers and fishing vessels from Vietnam, Indonesia, the Philippines, and China have occurred frequently in recent years [6]. The emergence of fishery incidents is one of the most crucial maritime threats, resulting in constant conflicts among states. For example, from the end of 2014 to April 2016, Indonesia seized 153 fishing boats from its maritime neighbors for "illegal fishing". Among them, 50 were from Vietnam, 43 were from the Philippines, and one was from China [7]. Another example is that, on 21 May 2017, Vietnam and Indonesia had a confrontation in the waters off the Natuna Islands, which Indonesia claims to be its exclusive economic zone. Indonesian patrol boats intercepted five Vietnamese fishing boats that had intruded into their exclusive economic zones and detained 11 Vietnamese fishers following their own laws [8]. In 2019, a Vietnamese fishing boat was sunk by a Chinese coast guard vessel near the disputed Paracel Islands, sparking protests from Vietnam [9]. These incidents highlight the ongoing tensions and conflicts in the South China Sea over fishing rights and territorial disputes. The lack of a comprehensive and effective framework for managing these conflicts has resulted in frequent incidents and escalated tensions among the countries involved [10].

The South China Sea is a region of great geopolitical importance due to its strategic location, rich natural resources, and overlapping maritime claims by several countries. IUU fishing is a significant problem in the region, and the involvement of external powers can complicate efforts to address it. One way in which external powers can complicate the issue of IUU fishing in the South China Sea is by supporting the maritime claims of one or more countries in the region. Doing so can exacerbate tensions and lead to an increase in IUU fishing activities, as countries may seek to assert their dominance over disputed waters. External powers can also contribute to IUU fishing in the region by providing financial or technical support to fishing activities. For example, subsidies provided by external

powers to their fishing companies may encourage overfishing and IUU fishing activities. Conflicting interests among external powers can also make it challenging to coordinate efforts to address IUU fishing in the South China Sea. For example, some countries may prioritize environmental concerns and sustainable fishing practices, while others may prioritize their security interests in the region. Therefore, addressing IUU fishing effectively in the South China Sea will require coordinated efforts and cooperation among countries and stakeholders.

*2.3. Disputes over Sovereignty and IUU Fishing*

The ongoing disputes over sovereignty in the South China Sea have significant impacts on IUU fishing in the region. These disputes create uncertainty and instability in the waters, making it difficult for countries to effectively manage their fisheries and prevent IUU fishing activities. Sovereignty disputes typically arise when neighboring countries lay claim to overlapping maritime areas, such as exclusive economic zones (EEZs) or continental shelves [11]. In the South China Sea, several countries have made overlapping maritime claims, including China, Vietnam, the Philippines, Malaysia, and Brunei. China has made a sweeping claim over most of the South China Sea, including areas that fall within the EEZs of other countries [12]. This conflict is due to differences in opinion on the South China Sea claims. China's claim has the nine-dash line, a map that outlines its historical claims to the region, as its historical background, and this line was first drawn by the Chinese government in the 1940s and has been a source of tension between China and its neighboring countries [13]. In contrast, other countries use a geographical location that refers to the International Law of the Sea Convention (UNCLOS), and they claim an exclusive economic zone (EEZ) extending 200 nautical miles (370 km) from their coasts and a continental shelf beyond that under the UNCLOS.

At the same time, the more deadlocked that the demarcation process is, the more likely that it is that the littoral states are to "increase their presence" in the South China Sea to strengthen their so-called sovereignty [14]. In this situation, fishing is endowed with distinct political colors; not only are fishing disputes in the South China Sea over fishery resources, but they also represent the political game between coastal states [15]. The fishing and sovereignty of the South China Sea have been firmly combined. From this perspective, the slow process of maritime delimitation in the South China Sea seriously hinders substantive solutions to the problem of IUU fishing.

In summary, due to the decrease in fishery resources and increase in demand, fishery disputes are escalating, further aggravating maritime conflicts over the sovereignty of the South China Sea, increasing the importance of the geopolitical situation in the South China Sea [10]. Currently, there is no effective regional cooperation in the conservation of fishery resources in the South China Sea, and IUU fishing behavior and fishery conflicts are constantly occurring. If coastal states do not undertake appropriate action in the South China Sea, marine biodiversity will be lost, and fishery resources may be depleted shortly thereafter [5]. In the states around the South China Sea, this depletion will be a fatal blow to fishery businesses and people's livelihoods and will cause serious damage to the marine environment of the South China Sea and the world.

## 3. How to Identify IUU Fishing

*3.1. Definition Standards*

3.1.1. Definition in IPOA-IUU

To define IUU fishing in the South China Sea, the current definition by international society should first be examined. It is predominantly defined in Article 3 of the International Plan of Action to Prevent, Deter and Eliminate Illegal, Unreported, and Unregulated Fishing (IPOA-IUU), which was developed by member states of the Food and Agriculture Organization of the United Nations (FAO) [16]. It mainly offers the definitions of the three fishing activities in question—"illegal," "unreported," and "unregulated"—as well as other terms used in the instrument. Although it is a soft law, the objective of the IPOA is to

"provide all states with comprehensive, effective transparent measures by which to act, including through appropriate regional fisheries management organizations established in accordance with international law."

IUU fishing is an integrated activity, not a specific activity [17]. Article 3.1 of the IPOA-IUU defines the basic elements of illegal fishing: "contravention of a state's laws and regulations," "contravention of the conservation and management measures adopted by the organization which the flag states are joined, or international law," and "contravention of the conservation of national law or international obligation." Regarding unreported fishing activities, Clause 3.2.1 of the IPOA-IUU refers to those "which have not been reported, or have been misreported, to the relevant national authority, in contravention of national laws and regulations," and Clause 3.2.2 refers to those "which have not been reported or have been misreported, in contravention of the reporting procedures of that organization." As opposed to Clause 3.3.1, these clauses govern those countries that are members of Regional Fisheries Management Organizations (RFMOs), which are the international organizations regulating regional fishing activities on the high seas. Thus, unreported fishing is a special case of "illegal" fishing due to violations of national law. Unregulated fishing appears to be the inverse. Unregulated fishing activities can be legal or illegal if national laws are enacted or if relevant international obligations are applied.

Moreover, the IPOA-IUU does not list specific activities related to all three factors. However, this approach gives states a great deal of discretion in making decisions based on the IPOA-IUU. In other words, states can argue that the definition is in line with their different activities. The UNCLOS, the United Nations Fish Stocks Agreement [13], the Compliance Agreement [18], and the FAO Code of Conduct [19] all consider and abide by the definition in the IPOA-IUU when identifying specific activities. Article 3 of the IPOA-IUU is an international guide currently used for the identification of IUU fishing.

### 3.1.2. Definitions in Different States

Apart from the IPOA-IUU and other international documents, there are also different interpretations from different entities that further complicate the issue.

To prevent, deter, and eliminate IUU fishing, the Philippines officially passed Republic Act No. 10,654 (Amending Philippine Fisheries Code of 1998) in 2014, including several new provisions that became the basis for an important shift in national legislation to combat IUU fishing. In Section 3, illegal fishing refers to fishing activities conducted by Philippine fishing vessels operating in violation of Philippine laws, Regional Fisheries Management Organization resolutions, and laws of other coastal states [20]. China's official documents provide multiple definitions and descriptions of IUU fishing. Article 2 of the Regulations on the Prevention and Control of Illegal, Unreported, and Unregulated Fishing in the People's Republic of China (referred to as the "IUU Fishing Regulations") defines "illegal, unreported, and unregulated fishing" as fishing activities conducted in violation of international, regional, or national laws, rules, and agreements. The China National Marine Fisheries Administration's "Guiding Opinions on Combating IUU Fishing" states that IUU fishing includes, but is not limited to, fishing in prohibited areas or times, using illegal fishing tools or methods, fishing for fish that do not meet specified standards, and failing to report or misreporting fishing activities. Overall, China's official definition is generally consistent with UNCLOS's definition.

In fact, compared with the definition from coastal states in the South China Sea, the European Union (EU) and US have undertaken the most far-reaching legislative efforts to specifically define and address IUU fishing. The EU has enacted Council Regulation (EC) No. 1005/2008 (referred to as the EU Regulation), adopting the scope and nature of the IPOA-IUU and creating a list of activities that can be considered IUU fishing. The EU IUU Regulation also states that a vessel is presumed to be engaged in IUU fishing if it transships or participates in joint fishing operations or supports or resupplies other vessels that are determined to be engaged in IUU fishing [21]. Overall, the EU IUU regulation has stronger legal force, more specific measures, stricter enforcement, and a traceability system

that make it more effective in combating IUU fishing than the IPOA-IUU. In 2007, the US adopted amendments to the Magnuson–Stevens Fishery Conservation and Management Reauthorization Act (MSA) on IUU fishing. The new Section 609 has two main components: a definition of IUU fishing and measures for the identification and listing of foreign nations with vessels that have engaged in IUU fishing. This definition further considers the impact of fishing activities on habitats and ecosystems, an area not discussed in the IPOA-IUU [21]. While the IPOA-IUU is an important international agreement that encourages sustainable fishing practices, the MSA is a more robust and comprehensive framework that provides stronger legal and scientific guidelines for managing and conserving fish populations.

### *3.2. Constitutive Elements of IUU Fishing*

Based on previous opinions, the IPOA-IUU specifies the scope of each component. When identifying IUU fishing in the South China Sea, based on the provisions of Article 3 of the IPOA-IUU, the following conditions should be met.

#### 3.2.1. Fishing Activities

(1)  Fishing activities: IUU fishing can be identified throughout the fishing process, which means that every step of fishing could be considered IUU fishing. Although Article 3.4 is an exception, the basic criterion of IUU fishing for the judgment of fishing behavior is whether the behavior complies with the laws, regulations, or international obligations with which the behavior should comply.

(2)  Behavior subject: vessels. The IPOA-IUU does not provide a definition of this term, but it is defined in the FAO Compliance Agreement as "any vessel used or intended for use for the purposes of the commercial exploitation of living marine resources, including mother ships and any other vessels directly engaged in such fishing operations" [18].

(3)  Behavior patterns: obtaining fishery resources. The current definition of the word "fishing" is provided in WTO fishery subsidy negotiation documents [22].

#### 3.2.2. The Law of the Place Where an Act Occurs Should Be Applicable to Identifying IUU Fishing

Regarding jurisdiction, Articles 3.1.1 and 3.2.1 of the IPOA-IUU refer to illegal and unreported fishing activities in the waters under the jurisdiction of a state, while Articles 3.1.2 and 3.2.2 refer to illegal and unreported fishing activities in the areas under the jurisdiction of the RFMOs. Within the purview of the RFMOs, Articles 3.1.2 and 3.1.3 of the IPOA-IUU apply to activities "conducted by vessels flying the flag of states that are parties to a relevant regional fisheries management organization" and "those undertaken by states cooperating with a regional fisheries management organization," respectively, limiting the scope of application. However, regardless of stipulations, the identification of fishing behavior is based on the nature of the water where the conduct occurs; that is, the place where fishing activities occur determines the choice and application of the law.

Whether based on the definition or elements of IUU fishing, identifying IUU behaviors is a complex and controversial issue. Factors such as the nature of the fishing area, the nature of the fishing vessel, the specific conduct, and the regional and international organizations to which the flag state belongs all need to be considered. In this process, the determinant is the nature of the water in which fishing occurs. However, it is clear that this issue is extremely controversial in the South China Sea, and identification must be performed more prudently.

## 4. China's Position and Measures on Combating IUU Fishing

### *4.1. China's Stance on Combating IUU Fishing*

#### 4.1.1. International Concerns about China

IUU fishing is a global issue, and China is one of the countries that has been criticized for engaging in this practice. Several international organizations and countries have expressed concerns about China's IUU fishing activities. The United Nations Food and

Agriculture Organization (FAO) has raised concerns about China's lack of effective measures to combat IUU fishing. In a 2016 report, the FAO noted that China was responsible for 20 percent of the world's fishing catch but that a significant portion of that catch was obtained through illegal, unreported, or unregulated means [23]. The European Union (EU) has also been critical of China's IUU fishing activities. In 2012, the EU issued a warning to China that it could face a ban on seafood imports if it did not undertake steps to address its IUU fishing practices [24]. In 2020, the EU renewed its warning and identified China as one of the top five countries engaged in IUU fishing [25].

4.1.2. China's Participation against IUU Fishing and the Sustainability of Global Fisheries

In addition, the international community has long argued that China has been unwilling to align its domestic policies with international rules on global fisheries governance [26]. China has been a major player in global fisheries governance and has been involved in various international agreements, such as the UNCLOS and the FAO Code of Conduct for Responsible Fisheries. However, there have been concerns among the international community that China has not fully aligned its domestic policies with these international rules, particularly in terms of sustainable fishing practices and IUU fishing. One issue that has been raised is China's large distant-water fishing fleet, which operates in waters around the world and has been accused of engaging in overfishing and IUU practices [27]. China has also been criticized for subsidizing its fishing industry, which could contribute to overfishing and distort global markets [28]. Additionally, there have been concerns about China's lack of transparency in reporting its fishing activities and the management of its fisheries [29]. There is still room for improvement, and the international community will continue to monitor China's progress in aligning its domestic policies with international rules in global fisheries governance.

However, as the world's largest fishing economy in terms of catch, production, and exports, China's actions to stop IUU fishing are important for the sustainable development of global fisheries. Moreover, China has undertaken steps to combat IUU fishing in recent years and has increased its participation in global efforts to address this issue. The Agreement on Port State Measures (PSMA) is the first binding international agreement to specifically target IUU fishing, aiming to prevent, deter, and eliminate IUU fishing by preventing vessels engaged in IUU fishing from using ports and landing their catches. China is a signatory to the PSMA and has taken steps to implement its provisions. In 2014, China enacted the "Regulations on the Administration of Port Inspection and Quarantine of Imported and Exported Aquatic Products," which are intended to help prevent IUU fishing by requiring that foreign vessels provide documentation demonstrating that their catch was obtained legally before being allowed to enter Chinese ports [30]. Overall, China's participation and cooperation against IUU fishing in the South China Sea have been increasing, and the country has shown a willingness to work with its neighbors and international organizations to promote sustainable fishing practices and preserve the marine environment.

*4.2. China Firmly Regards UNCLOS as Its Legal Basis in Combating IUU Fishing*

China's recent international norm-setting activities and domestic legislation regarding IUU fishing indicate a shift in Chinese policy from its previous reluctance to undertake action against IUU fishing. China hopes to improve its fisheries policy by updating its domestic fishing policies [31] and participating in international anti-IUU fishing negotiations.

IUU fishing directly endangers the sustainable development of fishery resources in the South China Sea; therefore, combating IUU fishing is an important task for all coastal states in the South China Sea. The United Nations Convention on the Law of the Sea (referred to as UNCLOS) specifically formulated the clause of "Conservation of living resources" for coastal states in Article 61, emphasizing that coastal states should adopt conservation and management measures to ensure that the living resources in exclusive economic zones (EEZs) are not endangered by over-exploitation. For instance, the coastal

state shall determine the allowable catch of the living resources in its EEZs, taking into account fishing patterns, the interdependence of stocks, any generally recommended international minimum standards, and available scientific information; and other data relevant to the conservation of fish stocks should be contributed and exchanged on a regular basis. Furthermore, according to Article 123 of the 1982 UNCLOS, the states of a semi-closed sea such as the SCS "should cooperate" with one another in the area of fisheries [32]. Therefore, China has undertaken active measures to control IUU fishing in the South China Sea, fulfilling its obligations under UNCLOS and international law.

### 4.3. China Is Implementing Measures to Keep Pace with the PSMA

The Agreement of Port State Measures (PSMA) was the first binding international agreement to specifically target IUU fishing [33]. The provisions of the PSMA can be applied to vessels trying to enter a designated port of a state that differs from their flag states. Its purpose is to restrain, deter, and reduce IUU fishing by preventing vessels from remaining in ports and unloading their catch. With this method, the PSMA limits the incentive for vessels engaged in IUU fishing to continue operation, keeping the fishery products of IUU fishing from entering domestic and international markets. The PSMA repeatedly emphasizes the value of the region, and unifying the various regions plays a basic role in the governance of global IUU fishing.

China signed the PSMA on 22 November 2016, indicating its intention to become a party to the treaty. However, China has not yet completed the ratification process by submitting the necessary instruments of ratification to the FAO. It is important to note that, even though China has not yet ratified the PSMA, it has undertaken steps to implement its provisions, including enacting domestic regulations to prevent IUU fishing and establishing a National Plan of Action to Combat IUU Fishing. Meanwhile, in the current measures against IUU fishing in the South China Sea, China has also made full reference to the ideas and measures in the PSMA. In addition to identifying IUU fishing vessels according to the IUU list provided by regional fishery organizations, some provisions of China's Fisheries Law (revised draft) include port state supervision of foreign fishing vessels [34]. At the same time, to implement the PSMA in the future, China has actively engaged in the staffing of law enforcement personnel, construction of port infrastructure, and capacity building of multi-sectoral cooperation.

### 4.4. Controlling IUU Fishing Aligns with the Idea of the 21st Century Maritime Silk Road Advocated by China

The 21st Century Maritime Silk Road is part of the Belt and Road Initiative (BRI) proposed by China [35]. The South China Sea is one of the most important areas in the "21st Century Maritime Silk Road." China proposes that states along the Belt and Road should jointly undertake the tasks of protecting the marine ecological environment and providing quality marine ecological services, with the goal of safeguarding global marine ecological security [36]. Based on the BRI, China adheres to three main principles in IUU fishing management: first, IUU fishing management should follow the premise of mutual respect; second, China will safeguard channels for dialogue and consultation, and maritime disputes will be resolved through dialogue, sincerity, and patience; third, states need to pursue win–win cooperation [37].

In this background, the pace of fishery cooperation at bilateral and multilateral levels between China and other neighboring states has been accelerated. For example, in 2013, China and Brunei signed the Joint Statement between the People's Republic of China and Brunei Darussalam. In 2018, China and Malaysia signed the Joint Statement, stating that the two states should continue to strengthen cooperation in fisheries. At the multilateral level, for example, the Declaration on the Coastal and Marine Environmental Protection of the South China Sea in the Next Ten Years (2017–2027), signed in 2017, stated that China and ASEAN should strengthen cooperation in fisheries, environmental protection, and ecology. Although China has signed a number of bilateral or multilateral agreements,

most agreements focus on the cooperation and coordination of fishery resources, and their provisions lack specific content [38]. The BRI would provide an opportunity for coastal states in the South China Sea to develop and implement fisheries governance frameworks and policies.

### 4.5. National Efforts to Control IUU Fishing

Effective prevention and reduction of IUU fishing in domestic fishery activities, including domestic fishers and large fishery enterprises, is necessary. In addition to economic sanctions, judicial and administrative instruments are required. China has undertaken considerable efforts to combat and address IUU fishing from a national perspective to reduce, control, and eliminate IUU fishing activities (See Table 1).

**Table 1.** National Regulations to Combat IUU Fishing.

| Regulation | Main Content | Nature | Time of Adoption |
|---|---|---|---|
| The Fisheries Law of the People's Republic of China (2013 Amendment) | Implementing a fishing quota system in accordance with the principle that the fishing amount shall be lower than the increasing amount of the fishery resources; a fishing license system; legal liability, such as fines, revoking of fishing licenses, confiscating of fishing vessels, and criminal liabilities. | Legally binding | 28 December 2013 |
| Revised Draft of the Fisheries Law for Comments (September 2019) | Enforcing regulations on IUU fishing; creating a 'blacklist' of IUU fishing in pelagic fisheries; reporting of fishing boats inbound and outbound; a fixed landing system for large- and medium-sized fishing boats; and a ban on foreign IUU fishing boats from entering Chinese ports. | Legally binding | Estimated take effect in 2023 |
| Detailed Rules for the Implementation of the Fisheries Law of the People's Republic of China (2020 Second Revision) | Improving fishery supervision and administration; tightening circumstances of fishing licenses; detailing and aggravating punishments. | Legally binding | 29 November 2020 |
| Measures of the People's Republic of China on the Registration of Fishing Vessels (2019 Amendment) | To strengthen the supervision and administration of fishing vessels; determine the ownership, nationality, port of registry, and other relevant legal relations of fishing vessels; and safeguard the legitimate rights and interests of all parties involved in fishing vessel registration. | Legally binding | 25 April 2019 |
| Provisions for the Administration of Pelagic Fishery | Prohibiting pelagic fishing enterprises, vessels, and ships engaging in illegal fishing. | Legally binding | 1 April 2020 |
| Measures for Monitoring the Location of Ocean-going Fishing Vessels (Revised Version) | To support the requirements of RFOs, when regional fishery organizations of which China is a member have stricter regulations on ship position monitoring, Chinese vessels shall abide by and implement the stricter regulations. | Legally binding | 2019 |

The existing domestic measures of IUU fishing are scattered among legal documents of various ranks in China, including the Fisheries Law of the People's Republic of China (2013 Amendment), Detailed Rules for the Implementation of the Fisheries Law of the

People's Republic of China (2020 Second Revision), and Measures of the People's Republic of China on the Registration of Fishing Vessels (2019 Amendment). Overall, a legal system for fishery governance with Chinese characteristics, based on the Fisheries Law and supplemented by various laws, regulations, rules, and international treaties, has taken shape.

The Fisheries Law was amended in 2013. Since 2019, substantial amendments to the Fisheries Law have been proposed and initiated. The Revised Draft for Comments was completed in September 2019, and it has made great progress in controlling IUU fishing in China. It is estimated that the revised Fisheries Law will take effect as soon as possible. The revised draft will tighten regulations on IUU fishing; meanwhile, the "blacklist" of IUU fishing in pelagic fisheries, the reporting of fishing boats inbound and outbound, a fixed landing system for large- and medium-sized fishing boats, and a ban on foreign IUU fishing boats from entering Chinese ports are all included in the revised draft. If the revised draft is formally implemented, Chinese fishing vessels that seriously violate the provisions could be "confiscated from their fishing vessels, suspended or canceled as deep-sea fishing enterprises, and the persons responsible put on a blacklist of deep-sea fishing employees" according to the law. Several provisions in the revised draft have been prepared for the implementation of the PSAM, for example, establishing procedures for the inspection of foreign vessels in non-fishing ports. The revised draft improves the legal effectiveness of the system to combat IUU fishing, reflecting the importance that China attaches to fishery resources; however, it could be more in line with relevant international treaties.

Simultaneously, support for the regulations of the Fisheries Law has developed. On 1 April 2020, provisions for the administration of pelagic fisheries came into force, and pelagic fishing enterprises, vessels, and ships were explicitly prohibited from engaging in IUU fishing activities [39]. Supporting increasingly stringent requirements for the management of fishing vessels from regional fishing organizations, at the end of August, China issued a revised version of the measures for monitoring the location of oceangoing fishing vessels, which came into force in 2019 [40]. According to these regulations, when regional fishery organizations in which China is a member have stricter regulations on ship position monitoring, Chinese ships shall abide by and implement stricter regulations, and fishing boats that dismantle or close their position monitoring systems without authorization will have their fishery subsidies deducted for that year [41].

## 5. Conclusions

IUU fishing poses a significant threat to fishery management, food security, state interests, and social stability. However, because the identification of IUU fishing is complex, and the causes of IUU fishing include many controversial issues, eliminating IUU fishing has a long way to go. In this situation, China's measures against IUU fishing have been criticized, but China's efforts to control IUU fishing, especially in the South China Sea, cannot be neglected [42]. Apart from adopting efficient measures from national and international perspectives to prevent IUU fishing, various forms of fisheries cooperation have been undertaken at the bilateral level. Simultaneously, China has undertaken efforts to conclude treaties in the region and actively refers to and intends to join the APSM. Certainly, some negative factors, such as maritime disputes, lack of trust, and weak enforcement, are challenging for China in establishing effective international cooperation in combating IUU fishing in the South China Sea. Addressing these challenges will require efforts to build trust among countries, address economic concerns, and strengthen enforcement measures. For example, joining SEAFDEC could be a positive step for China toward achieving this goal. In addition, the Center for Strategic and International Studies (CSIS) fisheries blueprint proposes several measures for China to improve the management of its fisheries resources, including strengthening its enforcement capabilities, improving data collection and sharing, promoting sustainable fishing practices, and enhancing international cooperation. Adopting these measures could help China to address the challenges that it faces in managing its vast and complex fisheries sector. Overall, China's participation and further efforts would play key roles against IUU fishing in the South China Sea and could

help to ensure that fishery resources are managed in a sustainable and equitable way for future generations.

**Author Contributions:** Writing—original draft, C.Y.; Writing—review & editing, Y.-C.C. All authors have read and agreed to the published version of the manuscript.

**Funding:** The fieldwork was supported by the following project: General Base Project of Liaoning Province Economic and Social Development Research, Research on Legal Basis for Construction of Maritime Capital City in Liaoning Province, China, Grant No. 2023lsljdybkt-004.

**Institutional Review Board Statement:** Not applicable.

**Informed Consent Statement:** Not applicable.

**Data Availability Statement:** Not applicable.

**Conflicts of Interest:** The authors declare no conflict of interest.

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
