# Peer review of "China’s Incentives and Efforts against IUU Fishing in the South China Sea"

_sustainability, doi:10.3390/su15097255_

Round 1

Reviewer 1 Report

This paper is well-written and logically expressed. The authors try to discuss the Chinese government's improvements and conditions in dealing with IUU fishing and building cooperation with international society. The paper is of research value and practical significance. I do, however, have some comments to improve the overall quality of the paper.

1. The introduction provides a clear and concise overview of the importance of fisheries industries in the South China Sea and the impacts of IUU fishing on the sustainable fishery. However, it is recommended to explain why the topic is important and relevant, and to give the reader an idea of what they can expect to read in the rest of the article.

2. The third part explores how to define IUU fishing, and in “3.1.2. Definition in Different states” authors provide Kuwait, European Union and the US. It is supposed to add China’s definition in this section, which would be more logical and relevant.

3. In the fourth part, "4. China's Position and Measures on Combating IUU Fishing," the author provides valuable points. It is recommended to put subheadings in a more academic and logical way.

4. Language, adding more linking or transition words that would signpost your arguments and can help to clarify your views.

Based on the above observations, I would recommend accepting this paper with minor revision.

Author Response

The authors would like to take this opportunity to thanks Editor-in-Chief and reviewers’ comments on this manuscript. The authors wish to address these comments as fully as possible and do hope that this paper is now reaching publishable level.

Reviewer #1:

  1. The introduction provides a clear and concise overview of the importance of fisheries industries in the South China Sea and the impacts of IUU fishing on the sustainable fishery. However, it is recommended to explain why the topic is important and relevant, and to give the reader an idea of what they can expect to read in the rest of the article.

Response: Many thanks for your comments. The introduction section has been reedited and added more research significance in this section at 55-58.

  1. The third part explores how to define IUU fishing, and in “3.1.2. Definition in Different states” authors provide Kuwait, European Union and the US. It is supposed to add China’s definition in this section, which would be more logical and relevant.

Response: Many thanks for your comments. The definition of China has been added at 206-218.

  1. In the fourth part, "4. China's Position and Measures on Combating IUU Fishing," the author provides valuable points. It is recommended to put subheadings in a more academic and logical way.

Response: Many thanks for your comments. Some subheading is this part has been revised.

  1. Language, adding more linking or transition words that would signpost your arguments and can help to clarify your views.

Response: Many thanks for your comments. The author has added more linking or transition words. 

Reviewer 2 Report

The topic selection of this paper is novel and has strong timeliness and innovation. Given the complex situation in the South China Sea, the paper discusses China’s incentives and efforts against IUU Fishing in the South China Sea through a series of convictive arguments. The research content of the paper mainly focuses on “causes of IUU fishing in the South China Sea, how to Identify IUU fishing and China’s position and measures on combating IUU fishing”. By combing the reasons, results and identification of IUU in this disputed area systematically, the paper discusses China’s participation and effort for the sustainable development of global fisheries, which refutes the international community’s criticism of China. In view of the content, the paper has clear ideas, full explanation and rigorous discussion and is quite scientific and academic. To prove the positive measures on combating IUU fishing in the South China Sea, the argumentation of the paper analyzes the cause of IUU fishing and identifies its scope and nature. Also, the paper lists detailed and sufficient information to illustrate the considerable efforts of China to reduce, control and eliminate IUU fishing activities. Obviously, the paper has a complete structure and logical level, with strong application value and readability. The authors pay attention to official reference sources, including domestic regulations, international legal instruments and data from international organizations. Hence, the references of the paper are closely related to the subject, with a high level and wide coverage. In brief, the paper is suggested to be accepted.

Author Response

Reviewer #2:

The topic selection of this paper is novel and has strong timeliness and innovation. Given the complex situation in the South China Sea, the paper discusses China’s incentives and efforts against IUU Fishing in the South China Sea through a series of convictive arguments. The research content of the paper mainly focuses on “causes of IUU fishing in the South China Sea, how to Identify IUU fishing and China’s position and measures on combating IUU fishing”. By combing the reasons, results and identification of IUU in this disputed area systematically, the paper discusses China’s participation and effort for the sustainable development of global fisheries, which refutes the international community’s criticism of China. In view of the content, the paper has clear ideas, full explanation and rigorous discussion and is quite scientific and academic. To prove the positive measures on combating IUU fishing in the South China Sea, the argumentation of the paper analyzes the cause of IUU fishing and identifies its scope and nature. Also, the paper lists detailed and sufficient information to illustrate the considerable efforts of China to reduce, control and eliminate IUU fishing activities. Obviously, the paper has a complete structure and logical level, with strong application value and readability. The authors pay attention to official reference sources, including domestic regulations, international legal instruments and data from international organizations. Hence, the references of the paper are closely related to the subject, with a high level and wide coverage. In brief, the paper is suggested to be accepted.

Response: Many thanks for your comments.  

Reviewer 3 Report

IUU fishing is not "becoming a global problem" - it's already a massive problem!

You can't have "conflicts among coastal states caused by IUU fishing". If it's IUU fishing then it's a dispute between a state and a non-state actor. If it's a conflict among states then it's a boundary dispute, not IUU

You can delete the Introduction section. It adds nothing to the paper.

The technical term is not "half-closed sea" but "semi-enclosed sea"

Can you explain how "the involvement of external powers" "complicates the issue of IUU fishing"? Helping coastal states to enforce their UNCLOS rights should be a major step towards preventing IUU fishing.

"Most claims are based on historical grounds related to the states involved" - you need to differentiate between territorial claims to islands and maritime claims to areas of sea.

This is incorrect - "For instance, Vietnam claims exclusive rights to its territory within 200 nautical miles of its coast." The word 'territory' is wrong here. Vietnam claims maritime rights based on UNCLOS.

When you say this "China claims 110 fishing rights in the South China Sea that extend beyond 200 nautical miles" you need some evidence that China claims this - what is the source? and you should say that this is a violation of UNCLOS.

What does this mean? "there is no legal basis to exercise jurisdiction over this gray zone of overlapping areas from the perspective of international law" - states are supposed to resolve these disputes through negotiation and arbitration.

You need to put UNCLOS at the front of your discussion. It provides a means to resolve inter-state disputes but China refuses to abide by its terms. Why?  

You mention "Regional Fisheries Management Organizations (RFMOs)" - why doesn't China join the existing SEAFDEC?

You say "China is a signatory to the PSMA agreement" but you don't mention  in this paragraph that China has NOT RATIFIED the agreement so it has no legal force in China.

You say "the Natuna Sea, where Indonesia has had longstanding disputes with China over territorial claims". There is no dispute over TERRITORIAL claims here. The dispute is about China claiming MARITIME ZONES beyond those allowed by UNCLOS.

"It demonstrates China's commitment to working with other nations to combat illegal fishing activities and promote sustainable fisheries management." One patrol in 2016 demonstrates nothing. If China was serious  then it would take action against its fishing vessels participating in IUU fishing inside Indonesia's EEZ.

This is not fully true "China has taken active measures to control IUU fishing in the South China Sea". China actively promotes IUU fishing by - for example - subsidising its fishing fleet to operate in other countries EEZs.

You say "China proposes that states along the Belt and Road should jointly undertake the tasks of protecting the marine ecological environment" - other countries don't take China seriously when it is actively promoting activity that undermines UNCLOS and maritime conservation.

"First, IUU fishing management should follow the premise of mutual respect" - NO, it should follow UNCLOS!

This is just wrong “China was the first to discover, name, develop, and utilize the South China Sea islands and related waters, and the first to start and continuously, peacefully, and effectively administer them". Please see the following for a better narrative: https://journals.sagepub.com/doi/abs/10.1177/0097700418771678

"states need to pursue win–win cooperation" - states also need to abide by the treaties they have ratified!

You say "IUU fishing poses a significant threat to fishery management, food security, state interests, and social stability." but a bigger threat is China's state-backed violations of UNCLOS that prevent regional fisheries cooperation. You need to mention this if your article is to have any value.

"China has attached great importance to the guiding significance of UNCLOS in combating IUU fishing in the South China Sea" - NO IT HASN'T! It has derided and ignored UNCLOS.

This analysis needs to separate itself from the Chinese government's position and take an independent look at the fisheries problems in the South China Sea.

Author Response

Reviewer #3:

  1. IUU fishing is not "becoming a global problem" - it's already a massive problem!

Response: Many thanks for your comments. This sentence has been changed to “ Illegal, unreported, and unregulated (IUU) fishing is a massive problem that poses a significant threat to the sustainability of marine ecosystems and the livelihoods of millions of people who depend on fishing for their food and income.”

  1. You can't have "conflicts among coastal states caused by IUU fishing". If it's IUU fishing then it's a dispute between a state and a non-state actor. If it's a conflict among states then it's a boundary dispute, not IUU.

Response: Many thanks for your comments. “Conflicts among coastal states caused by IUU fishing” has been deleted.

  1. You can delete the Introduction section. It adds nothing to the paper.

Response: Many thanks for your comments. The introduction section has been reedited.

  1. The technical term is not "half-closed sea" but "semi-enclosed sea"

Response: Many thanks for your comments. It has been revised.

  1. Can you explain how "the involvement of external powers" "complicates the issue of IUU fishing"?

Response: Many thanks for your comments. One way that external powers can complicate the issue of IUU fishing in the South China Sea is by supporting the territorial claims of one or more countries in the region. This can exacerbate tensions and lead to an increase in IUU fishing activities as countries may seek to assert their dominance over disputed waters. External powers can also contribute to IUU fishing in the region by providing financial or technical support to fishing activities. Conflicting interests among external powers can also make it challenging to coordinate efforts to address IUU fishing in the South China Sea. Therefore, addressing IUU fishing effectively in the South China Sea will require coordinated efforts and cooperation among countries and stakeholders.

  1. Helping coastal states to enforce their UNCLOS rights should be a major step towards preventing IUU fishing.

Response: Many thanks for your comments.

  1. "Most claims are based on historical grounds related to the states involved" - you need to differentiate between territorial claims to islands and maritime claims to areas of sea.

Response: Many thanks for your comments. This part has been rewritten and tried to be more correct.

  1. This is incorrect - "For instance, Vietnam claims exclusive rights to its territory within 200 nautical miles of its coast." The word 'territory' is wrong here. Vietnam claims maritime rights based on UNCLOS.

Response: Many thanks for your comments. This part has been revised as “For instance, Vietnam claims an exclusive economic zone (EEZ) extending 200 nautical miles (370 kilometers) from its coast and a continental shelf beyond that under the United Nations Convention on the Law of the Sea (UNCLOS).”

  1. When you say this "China claims 110 fishing rights in the South China Sea that extend beyond 200 nautical miles" you need some evidence that China claims this - what is the source? and you should say that this is a violation of UNCLOS.

Response: Many thanks for your comments. This part has been rewritten.

  1. What does this mean? "there is no legal basis to exercise jurisdiction over this gray zone of overlapping areas from the perspective of international law" - states are supposed to resolve these disputes through negotiation and arbitration.

Response: Many thanks for your comments. This part has been rewritten

  1. You need to put UNCLOS at the front of your discussion. It provides a means to resolve inter-state disputes but China refuses to abide by its terms. Why?  

Response: Many thanks for your comments. This article has put UNCLOS at a important discussion. Whether China fully complied with UNCLOS or not, possibly due to considerations regarding territorial disputes, but this does not affect China's efforts to govern IUU fishing.

  1. You mention "Regional Fisheries Management Organizations (RFMOs)" - why doesn't China join the existing SEAFDEC?

Response: Many thanks for your comments. “Why doesn't China join the existing SEAFDEC” has many reasons and it not discussed in this article.

  1. You say "China is a signatory to the PSMA agreement" but you don't mention in this paragraph that China has NOT RATIFIED the agreement so it has no legal force in China.

Response: Many thanks for your comments. But “China has NOT RATIFIED the agreement” is at 358-359.

  1. You say "the Natuna Sea, where Indonesia has had longstanding disputes with China over territorial claims". There is no dispute over TERRITORIAL claims here. The dispute is about China claiming MARITIME ZONES beyond those allowed by UNCLOS.

Response: Many thanks for your comments. It has been revised.

  1. "It demonstrates China's commitment to working with other nations to combat illegal fishing activities and promote sustainable fisheries management." One patrol in 2016 demonstrates nothing. If China was serious then it would take action against its fishing vessels participating in IUU fishing inside Indonesia's EEZ.

This is not fully true "China has taken active measures to control IUU fishing in the South China Sea". China actively promotes IUU fishing by - for example - subsidising its fishing fleet to operate in other countries EEZs.

You say "China proposes that states along the Belt and Road should jointly undertake the tasks of protecting the marine ecological environment" - other countries don't take China seriously when it is actively promoting activity that undermines UNCLOS and maritime conservation.

"First, IUU fishing management should follow the premise of mutual respect" - NO"states need to pursue win–win cooperation" - states also need to abide by the treaties they have ratified!

You say "IUU fishing poses a significant threat to fishery management, food security, state interests, and social stability." but a bigger threat is China's state-backed violations of UNCLOS that prevent regional fisheries cooperation. You need to mention this if your article is to have any value.

"China has attached great importance to the guiding significance of UNCLOS in combating IUU fishing in the South China Sea" - NO IT HASN'T! It has derided and ignored UNCLOS. it should follow UNCLOS!

Response: Many thanks for your comments.

  1. This is just wrong “China was the first to discover, name, develop, and utilize the South China Sea islands and related waters, and the first to start and continuously, peacefully, and effectively administer them". Please see the following for a better narrative:https://journals.sagepub.com/doi/abs/10.1177/0097700418771678

Response: Many thanks for your comments. It has been deleted.

  1. This analysis needs to separate itself from the Chinese government's position and take an independent look at the fisheries problems in the South China Sea.

Response: Many thanks for your comments. 

Round 2

Reviewer 3 Report

"One way that external powers can complicate the issue of IUU fishing in the South China Sea is by supporting the territorial claims of one or more countries in the region." - which external powers support TERRITORIAL claims? None do. What they support are MARITIME claims based upon UNCLOS - which China has also ratified.

"Vietnam claims an exclusive economic zone (EEZ)" - you should also include the Philippines, Malaysia, Brunei and Indonesia here. In fact all states around the South China Sea claim an EEZ based upon UNCLOS - except China. China has never formally stated the legal basis of its claim to the South China Sea fisheries.

This is overstated: "there is no legal basis to exercise jurisdiction over this gray zone of overlapping areas from the perspective of international law which means there is no clear and undisputed legal authority for any country or jurisdiction to govern this area or activity according to international law" Yes, there is. It is called UNCLOS. What is missing is a single body to judge these disputes but they can be judged by the ICJ, ITLOS or a tribunal.

You should say that the IPOA-IUU was developed by member states of the FAO.

What is the source of this quote "“provide all states with comprehensive, effective transparent measures by which to act, including through appropriate regional fisheries management organizations established in accordance with international law."

Line 196 - why are you talking about Kuwait? It's not in the South China Sea!

Line 215 - why are you talking about the EU here?

Line 226 - why are you talking about the US here?

Line 256 - you need to tell us what an RFMO is here

Line 287 - is there a word missing here? "China’s Participation in against IUU Fishing" 

Line 309 - you need to introduce the PSMA agreement here

Line 322 "China's participation in joint patrols and inspections with other countries is seen as a positive step in the fight against IUU fishing." - Seen by whom? There's no source here. This sounds like Chinese government propaganda.

Line 324 "It demonstrates China's commitment to working with other nations to combat illegal fishing activities and promote sustainable fisheries management." No it doesn't. One patrol in November 2016 sounds pathetic!

Section 4.4 reads like Chinese government propaganda, particularly this sentence: "The littoral states of the South China Sea should respect the historic rights of China." What historic rights are these? Why should other states respect them? The point of ratifying UNCLOS is that states give up any so-called historic rights. See the attached paper which explains how China made-up its historic rights claim in the early 1990s.

"China will also consider the reasonable fishing needs and rights of neighboring states in the South China Sea, which demonstrates its respect for UNCLOS" What is the evidence for this? Can you point to any government law or statements that support this? What China should do is announce its commitment to UNCLOS and to EEZs.

"The history of co-operation between China and littoral states in the South China Sea shows that overall cooperation in this area brings far more opportunities and benefits to littoral states than conflict and loss." What examples can you provide for this giant claim?

Author Response

Response to Editor and Reviewers’ Comments-Second Round

The authors would like to take this opportunity to thanks Editor-in-Chief and reviewers’ comments on this manuscript. The authors wish to address these comments as fully as possible and do hope that this paper is now reaching publishable level.

Reviewer #4:

  1. One way that external powers can complicate the issue of IUU fishing in the South China Sea is by supporting the territorial claims of one or more countries in the region." - which external powers support TERRITORIAL claims? None do. What they support are MARITIME claims based upon UNCLOS - which China has also ratified.

Response: Many thanks for your comments. The “territorial” has been revised.

  1. "Vietnam claims an exclusive economic zone (EEZ)" - you should also include the Philippines, Malaysia, Brunei and Indonesia here. In fact all states around the South China Sea claim an EEZ based upon UNCLOS - except China. China has never formally stated the legal basis of its claim to the South China Sea fisheries.

Response: Many thanks for your comments. This has been changed to “While other countries use a geographical location that  refers to the International Law of the Sea Convention(UNCLOS), and they claim an exclusive economic zone (EEZ) extending 200 nautical miles (370 kilometers) from its coast and a continental shelf beyond that under the UNCLOS. ”

  1. This is overstated: "there is no legal basis to exercise jurisdiction over this gray zone of overlapping areas from the perspective of international law which means there is no clear and undisputed legal authority for any country or jurisdiction to govern this area or activity according to international law" Yes, there is. It is called UNCLOS. What is missing is a single body to judge these disputes but they can be judged by the ICJ, ITLOS or a tribunal.

Response: Many thanks for your comments. This appropriate description has been deleted.

  1. You should say that the IPOA-IUU was developed by member states of the FAO.

Response: Many thanks for your comments. It has been added.

  1. What is the source of this quote "“provide all states with comprehensive, effective transparent measures by which to act, including through appropriate regional fisheries management organizations established in accordance with international law."

Response: Many thanks for your comments. It is from the International Plan of Action to Prevent, Deter and Eliminate Illegal, Unreported and Unregulated Fishing (IPOA-IUU).

  1. Line 196 - why are you talking about Kuwait? It's not in the South China Sea!

The example of Kuwait is really not pertinent and I have deleted that part.

  1. Line 215 - why are you talking about the EU here? Line 226 - why are you talking about the US here?

Many thanks for your comments. And comparison was added in this section. 

  1. Line 256 - you need to tell us what an RFMO is here.

Many thanks for your comments. It has been added at 178-179.

  1. Line 287 - is there a word missing here? "China’s Participation in against IUU Fishing" 

Many thanks for your comments. It has been revised.

  1. Line 309 - you need to introduce the PSMA agreement here.

Many thanks for your comments. It has been added.

  1. Line 322 "China's participation in joint patrols and inspections with other countries is seen as a positive step in the fight against IUU fishing." - Seen by whom? There's no source here. This sounds like Chinese government propaganda.

Line 324 "It demonstrates China's commitment to working with other nations to combat illegal fishing activities and promote sustainable fisheries management." No it doesn't. One patrol in November 2016 sounds pathetic!

Many thanks for your comments. It has been revised.

  1. Section 4.4 reads like Chinese government propaganda, particularly this sentence: "The littoral states of the South China Sea should respect the historic rights of China." What historic rights are these? Why should other states respect them? The point of ratifying UNCLOS is that states give up any so-called historic rights. See the attached paper which explains how China made-up its historic rights claim in the early 1990s.
  2. "China will also consider the reasonable fishing needs and rights of neighboring states in the South China Sea, which demonstrates its respect for UNCLOS" What is the evidence for this? Can you point to any government law or statements that support this? What China should do is announce its commitment to UNCLOS and to EEZs.
  3. "The history of co-operation between China and littoral states in the South China Sea shows that overall cooperation in this area brings far more opportunities and benefits to littoral states than conflict and loss." What examples can you provide for this giant claim?

Many thanks for your comments. The inappropriate statements have been deleted and this section has also been rewritten, aiming to be more academic and reasoned.

Round 3

Reviewer 3 Report

This is much better. There are however, some problems on page 7.

Line 316 - "China has also participated in the Regional Code of Conduct in the South China Sea adopted by the member states of ASEAN and China."
No Code of Conduct has yet been agreed. There have been discussions about agreeing one for more than 25 years...

Line 318 - "Under this code, in 2011, China-ASEAN Maritime Cooperation Fund was initiated"
No, the Fund is not under the Code of Conduct. It is a unilateral measure by China.

Line 320 "which played an essential role in developing fisheries preservation in the South China Sea area"
Actually, I think the fund has done nothing at all in the field of fisheries preservation. What evidence do you have for this statement?

Line 321 "It demonstrates China's commitment to working with other states to combat illegal fishing activities and promote sustainable fisheries management."
Actually, the failure of the fund to do anything on this issue demonstrates the opposite. Why do you keep trying to insert these propaganda points into the article?

The reference you use is pure government propaganda, not scientific or accurate at all: C. Penghong, China-ASEAN Maritime Cooperation: Process, Motivation, and Prospects. China Int’l Stud. 2015, 53,26–40. https://www.ciis.org.cn/english/COMMENTARIES/202007/t20200715_2736.html

Line 395 "China has also participated in RFMOs" - which ones? Not in the South China Sea...

Line 462 "It is estimated that the revised Fisheries Law will take effect by 2022."
It's now 2023 - has it taken effect?

Line 498 "China has attached great importance to the guiding significance of UNCLOS in combating IUU fishing in the South China Sea"
Have you got any evidence to support this statement?

Why don't you make some brave recommendations in your paper? Here are some suggestions...

1. China should stick to UNCLOS rules for delimiting its EEZ in which to manage fish stocks.

2. China should join SEAFDEC and turn it into an RFMO for the South China Sea.

3. China should adopt the measures proposed in the CSIS fisheries blueprint
https://amti.csis.org/coc-blueprint-fisheries-environment/

All the best!

Author Response

Response to Editor and Reviewers’ Comments-Third Round

The authors would like to take this opportunity to thanks Editor-in-Chief and reviewers’ comments on this manuscript. The authors wish to address these comments as fully as possible and do hope that this paper is now reaching publishable level.

Reviewer #3:

  1. This is much better. There are however, some problems on page 7.

Line 316 - "China has also participated in the Regional Code of Conduct in the South China Sea adopted by the member states of ASEAN and China.”No Code of Conduct has yet been agreed. There have been discussions about agreeing one for more than 25 years...

Line 318 - "Under this code, in 2011, China-ASEAN Maritime Cooperation Fund was initiated" No, the Fund is not under the Code of Conduct. It is a unilateral measure by China.

Line 320 "which played an essential role in developing fisheries preservation in the South China Sea area"

Actually, I think the fund has done nothing at all in the field of fisheries preservation. What evidence do you have for this statement?

Line 321 "It demonstrates China's commitment to working with other states to combat illegal fishing activities and promote sustainable fisheries management."

Actually, the failure of the fund to do anything on this issue demonstrates the opposite. Why do you keep trying to insert these propaganda points into the article?

Response: Many thanks for your comments. As for this topic has some inappropriate statements, the author has deleted relevant discussion about “the Regional Code of Conduct”.

  1. The reference you use is pure government propaganda, not scientific or accurate at all: C. Penghong, China-ASEAN Maritime Cooperation: Process, Motivation, and Prospects. China Int’l Stud. 2015, 53,26–40. https://www.ciis.org.cn/english/COMMENTARIES/202007/t20200715_2736.html

Response: Many thanks for your comments. This reference has been deleted.

  1. Line 395 "China has also participated in RFMOs" - which ones? Not in the South China Sea...

Response: Many thanks for your comments. Yes, it is not in the South China Sea and this statement has been deleted.

  1. Line 462 "It is estimated that the revised Fisheries Law will take effect by 2022."

It's now 2023 - has it taken effect?

Response: Many thanks for your comments. It does not take effect yet, and this incorrect statement has been revised.

  1. Line 498 "China has attached great importance to the guiding significance of UNCLOS in combating IUU fishing in the South China Sea" Have you got any evidence to support this statement?

Response: Many thanks for your comments. This section has been revised, and this sentence has been deleted.

  1. Why don't you make some brave recommendations in your paper? Here are some suggestions... China should stick to UNCLOS rules for delimiting its EEZ in which to manage fish stocks. 2. China should join SEAFDEC and turn it into an RFMO for the South China Sea. 3. China should adopt the measures proposed in the CSIS fisheries blueprint https://amti.csis.org/coc-blueprint-fisheries-environment/

Response: Many thanks for your comments. Some recommendations have been added in the conclusion.
